# Tangent Transformers for Composition, Privacy and Removal

**Tian Yu Liu**\*
University of California, Los Angeles
tianyu@cs.ucla.edu

**Aditya Golatkar**\*
University of California, Los Angeles
adityagolatkar@ucla.edu

**Stefano Soatto**
University of California, Los Angeles
soatto@cs.ucla.edu

## ABSTRACT

We introduce Tangent Attention Fine-Tuning (TAFT), a method for fine-tuning linearized transformers obtained by computing a First-order Taylor Expansion around a pre-trained initialization. We show that the Jacobian-Vector Product resulting from linearization can be computed efficiently in a single forward pass, reducing training and inference cost to the same order of magnitude as its original non-linear counterpart, while using the same number of parameters. Furthermore, we show that, when applied to various downstream visual classification tasks, the resulting Tangent Transformer fine-tuned with TAFT can perform comparably with fine-tuning the original non-linear network. Since Tangent Transformers are linear with respect to the new set of weights, and the resulting fine-tuning loss is convex, we show that TAFT enjoys several advantages compared to non-linear fine-tuning when it comes to model composition, parallel training, machine unlearning, and differential privacy. Our code is available at: https://github.com/tianyu139/tangent-model-composition

## 1 INTRODUCTION

Deep Networks are highly non-linear operators trained by optimizing highly non-convex functions, yet some of the training dynamics near convergence approximate those of linear over-parameterized systems (Saxe et al., 2013). Accordingly, linearization has been used as a tool for both the analysis of deep networks (Jacot et al., 2018), and their design (Achille et al., 2021). Linearization around an initial set of weights, however, is of limited practical relevance since the early learning dynamics are highly non-linear and decisive of final performance (Golatkar et al., 2019). On the other hand, linearization around a pre-trained point has been shown to be essentially equivalent to non-linear fine-tuning, and better in the case of few-shot fine-tuning (Achille et al., 2021). A linearized model has the same number of parameters as the original, but carries some distinct advantages: First, linearity allows straightforward model composition, whereby ensemble models can be formed by scalar combinations at essentially zero cost. Second, a monolithic training set can be partitioned into smaller "shards," for instance for privacy or attribution purposes, and the resulting models combined to yield performance similar to a model trained on the monolith. This results in zero loss compartmentalization of separately trained models, and enables seamless parallel training. Third, since the linearized model can be trained by optimizing a convex loss, existing methods for private training via selective forgetting (Abadi et al., 2016) are effective and enjoy strong theoretical guarantees.

Despite the benefits, model linearization is challenging at scale. To this date, only small-scale models have been successfully shown to operate comparably to their non-linear counterparts, typically in the ResNet family of architectures. To our knowledge, our work is the first to propose an efficient method to linearize models in the Transformer family of architectures, leading to what we call "Tangent Transformers." Tangent Transformers can be used to adapt Transformer models, as an alternative to prompt-tuning, fine-tuning, or adapter training, none of which are linear in weight space.

---

\*Denotes equal contribution

The key to enable practical linearization of Transformers is an efficient way to compute the Jacobian-Vector product in a single forward pass, described in Sec. 3.2. As a result, training and inference costs are on the same order of magnitude as the corresponding non-linear Transformer model. In Sec. 4.2 we show that a Tangent Vision Transformer (T-ViT) can achieve similar accuracy to non-linear fine-tuning (NLFT) of the original ViT (Dosovitskiy et al., 2020) model. Given the comparable accuracy, we focus on illustrating some of the benefits of Tangent Transformers in Sec. 4. Specifically:

**Compositionality:** Linearity yields equivalence between composition in weight space and composition in activations, *i.e.*, ensembling. This allows seamlessly combining independently trained models, with obvious benefits to parallel, incremental, and federated learning, while maintaining a constant inference time compared to traditional ensembling.

**Speedup:** Specifically, we achieve up to $10\times$ $(50\times)$ speed-up in parallel training, with only $3.7\%(9.3\%)$ drop in overall accuracy compared to non-linear fine-tuning on the full dataset, improving over the Model Soup (Wortsman et al., 2022b) approach by $9.1\%(13.5\%)$ respectively.

**Compartmentalization:** Since training on disjoint shards yields the same performance, data removal, if it becomes necessary or desirable (Achille et al., 2023), can be performed deterministically in an exact fashion, at essentially zero cost.

**Privacy:** Most theoretical results and practical methods concerning Differential Privacy (DP) (Abadi et al., 2016; Bassily et al., 2014; Fang et al., 2023; Yang et al., 2022; Bassily et al., 2021; Wang et al., 2019; 2022a) provide much better utility-privacy trade-offs when the optimization problem being solved is convex. While in general deep networks are not, if pre-training is conducted on safe data, linearized fine-tuning is convex and therefore strong results and effective methods for DP apply.

In Sec. 2 we briefly survey relevant related work, and in Sec. 3 we derive our method for Tangent Attention Fine-Tuning (TAFT). We illustrate the benefits of TAFT in Sect. 3.2 for parallel training and composition, in Sec. 3.3 for selective forgetting, or "unlearning", and in Sec. 3.4 for privacy. Finally, we empirically evaluate TAFT in Sec. 4.

## 2 RELATED WORK

**Deep network linearization:** Deep networks linearized using the first-order taylor approximation have various interpretations in literature - viewing gradients as features (Mu et al., 2020), learning along the tangent space of a neural network (Liu et al., 2022; Liu & Soatto, 2023), infinite-width networks (Jacot et al., 2018). Mu et al. (2020) shows that the Jacobian-Vector product (JVP) of linearized convolutional networks can be efficient computed in a single modified forward pass. Achille et al. (2021) shows that by using Leaky-ReLU activations and training with the rescaled square loss (Hui & Belkin, 2020) and gradient pre-conditioning, ResNets linearized around ImageNet pre-trained weights can achieve comparable performances to the original non-linear networks on downstream fine-tuning tasks. Most similar to our work, Liu & Soatto (2023) applies linearized convolutional networks to ensembling and continual fine-tuning. To the best of our knowledge, we are the first work to linearize transformer networks in a manner that is both computationally efficient and achieves competitive results when fine-tuned on various downstream tasks.

**Composition:** We investigate compositionality of deep networks in weight space to yield a model that generalizes better than each individual component model. Weight averaging has been used to improve generalization of pre-trained weights (Choshen et al., 2022), and for distributed fine-tuning (Liu & Soatto, 2023; Wortsman et al., 2022a). Wortsman et al. (2022b) averages the weights of large pre-trained models fine-tuned with different hyperparameters to improve generalization. Compositionality has also been explored through prompts in continual learning (Wang et al., 2022b;c). However, these works do not develop any theoretically meaningful interpretations for composition, and often scale in inference time as number of component models increase. We introduce TAFT for linearly composing tangent transformer models trained, possibly in parallel, on multiple disjoint shards of data. Under our method, composition of linear weights is theoretically equivalent to output ensembling with constant inference cost.

**Machine Unlearning:** Machine unlearning, or forgetting, methods aim to remove the influence of specific samples from a trained network (Achille et al., 2023; Bourtoule et al., 2021; Dukler et al., 2023; Golatkar et al., 2021; 2020a;b). We focus on methods that are zero-shot and yields theoretically guaranteed unlearning. These works (Bourtoule et al., 2021; Bowman et al., 2023; Koch & Soll) operate by splitting datasets into multiple disjoint shards, hence compartmentalizing each sample to only a single shard. Unlearning can then be done by simply removing the shard. However, such

methods either incur high inference costs as a result of running inference across multiple models, and often incur significant trade-offs in generalization accuracy of the composed model. Instead, we show that as a result of linearity, we can compose tangent transformer networks simply by averaging network weights, to produce outputs that are equivalent to the ensemble of each network with an inference cost that is constant with respect to number of shards/models in the ensemble.

**Privacy:** Differential privacy (Dwork et al., 2014) seeks to limit the amount of information a trained model contains about the individual training samples. DP-SGD (Abadi et al., 2016) achieves this through clipping of individual gradients followed by addition of Gaussian noise. Bassily et al. (2021; 2014); Fang et al. (2023); Wang et al. (2019; 2022a); Yang et al. (2022) provide rigorous theoretical guarantees for the convergence and utility of DP algorithms, and show that convex (or strongly convex) models offer better utility compared to their non-convex counterparts. Per dimension Gaussian noise in DP-SGD reduces the utility of training large models in favour of fine-tuning parameter efficient adapters (Bu et al., 2022b;a; Golatkar et al., 2022; Yu et al., 2021). In this paper, we show that TAFT in the tangent space of these parameters provides better utility-privacy trade-off.

## 3 METHOD

We explore the most direct way to linearize a pre-trained transformer network $f_w$ - by replacing it with its first-order taylor approximation $f_w^{lin}$ about its pre-trained weights $w$.

$$f_w^{lin}(\cdot) = f_w(\cdot) + \nabla_w f_w(\cdot) \cdot \Delta w \tag{1}$$

By construction, $f_w^{lin}$ is now linear with respect to $\Delta w$, the new set of learnable parameters.

The new network can be trained easily using any loss function. For example, using the standard mean-squared error loss yields a quadratic objective function, reducing the training of such models to simple linear-quadratic optimization (Achille et al., 2021). We use the Rescaled Square Loss (RSL) (Hui & Belkin, 2020) given by

$$L(x,y) = \frac{1}{K} \big( \alpha([f_w^{lin}(x)]_y - \kappa)^2 + \sum_{i=1,i\neq y}^{K} ([f_w^{lin}(x)]_i)^2 \big) \tag{2}$$

where $\alpha, \kappa$ are hyper-parameters. We empirically found RSL performs better compared to cross-entropy or regular MSE loss, corroborating the results of Achille et al. (2021); Liu & Soatto (2023).

We further note that how good a local approximation of the network is depends on the distance that the fine-tuned weight moves from its initial point $w$. As such, we additionally regularize the training objective by adding a penalty on $\|\Delta w\|_2^2$. The resulting training objective is simply ridge regression, retaining the benefits of linear-quadratic optimization while obtaining better empirical results (Sec. 4.6). Note that due to the high dimensionality of the training set and gradient-based features, it is computationally prohibitive to obtain the closed form solution even though it exists.

This appears costly to compute for both inference and training, since evaluating the Jacobian-Vector Product (JVP) $\nabla_w f_w(x) \cdot \Delta w$ requires computing the gradient with respect to the original weights, for every input $x$. However, by computing the directional derivative, we can derive closed form equations for the linearized versions of the key building blocks of transformer networks. We show that they can be computed in a single modified forward pass through the original model where each layer of the network outputs the computed JVP ($\mathrm{JVP}_{out}$) in addition to the original output values, and takes as input the JVP from the previous layer ($\mathrm{JVP}_{in}$) in addition to the original input values.

### 3.1 LINEARIZING TRANSFORMERS

Here, we will derive the closed form linearization of a transformer network, and show that it can be easily computed by the modified forward propagation without explicitly computing any gradients. We break down transformer networks into attention, normalization, and fully-connected layers, and separately derive their linearizations (note that while fully-connected layers are already linear, we still need to handle the input JVP from the previous layer). These layers can be simply composed together to form the final Tangent Transformer network.

We parameterize the attention function $A : \mathbb{R}^{d \times n} \mapsto \mathbb{R}^{d \times n}$ by the weights $W_q, W_k, W_v \in \mathbb{R}^{d \times d}$ corresponding to the key, query and value matrices respectively, and given by

$$A(x) = \Phi(x)V(x), \quad \text{where } \Phi(x) = \sigma(Q(x)K(x)^T), \tag{3}$$

$$Q(x) = \langle W_q, x \rangle, K(x) = \langle W_k, x \rangle, V(x) = \langle W_v, x \rangle \tag{4}$$

where $\sigma$ is the soft-max activation function. We will write $Q, K, V, \Phi$ instead of $Q(x), K(x), V(x), \Phi(x)$ for ease of notation. For simplicity, we only consider single-headed attention in our derivations, but note that our definitions and derivations can be extended to multi-headed attention (which we use in the experiments section) with minimal modification. Now, we wish to compute the first-order approximation of $A$, denoted $A_{lin} : \mathbb{R}^{d \times n} \mapsto \mathbb{R}^{d \times n}$, and parameterized by the linearized weights $\Delta W_q, \Delta W_k, \Delta W_v$ for the key, query, and value matrices respectively. By taking directional derivatives, we can derive the following closed form expression for $A_{lin}$ (details can be found in Appendix B):

$$A_{lin}(x) = A(x) + \underbrace{\lim_{r \to 0} \frac{\partial}{\partial r} A(x, W_q + r\Delta W_q, W_k + r\Delta W_k, W_v + r\Delta W_v)}_{\text{JVP}_{out}} \tag{5}$$

$$= A(x) + \left( \Phi \odot \Psi - (\mathbb{1} \odot (\Phi^T \Psi))\Phi \right)^T V + \Phi\Gamma \tag{6}$$

where

$$\Psi := \Psi(x) := \langle \Delta Q + W_q^T \, \text{JVP}_{in}, K \rangle + \langle Q, \Delta K + W_k^T \, \text{JVP}_{in} \rangle \tag{7}$$

$$\Gamma := \Gamma(x) := \Delta V + W_v^T \, \text{JVP}_{in} \tag{8}$$

$$\Delta Q := \langle \Delta W_q, x \rangle, \; \Delta K := \langle \Delta W_k, x \rangle, \; \Delta V := \langle \Delta W_v, \, x \rangle \tag{9}$$

$\odot$ denote the Hadamard product, and $\text{JVP}_{in} = \lim_{r \to 0} \frac{\partial x}{\partial r}$ is the Jacobian-Vector Product computed from the previous layer, obtained from the modified forward pass. The terms $\Phi, Q, K, V$ can be obtained for free as intermediate variables from computing $A(x)$. Thus, computing the JVP term is done through simple matrix multiplication operations of similar computational complexity as the original attention mechanism.

Transformer blocks also include several normalization layers. Similarly we can compute a closed form expression for their linearized versions that can be obtained in the modified forward propagation step. We show the derivation for Layer Norm, which we denote $LN_{(\gamma,\beta)}(\cdot)$ and parameterize by the affine transformation parameters $(\gamma, \beta)$, but note that the results can be easily generalized to other forms such as Batch Norm (Achille et al., 2021). In particular, the linearized Layer Norm $LN_{lin}$, which is parameterized by $(\Delta\gamma, \Delta\beta)$, evaluated at $x$ can be computed as

$$LN_{lin}(x) = LN_{(\gamma,\beta)}(x) + \underbrace{LN_{(\Delta\gamma,\Delta\beta)}(x)}_{} + \tag{10}$$

$$\underbrace{\frac{1}{\sqrt{Var[x]}} \left( (\text{JVP}_{in} - \mathbb{E}[\text{JVP}_{in}]) - \frac{\mathbb{E}[(x - \mathbb{E}[x])(\text{JVP}_{in} - \mathbb{E}[\text{JVP}_{in}])] \cdot (x - \mathbb{E}[x])}{Var[x]} \right) * \gamma}_{\text{JVP}_{out}}$$

$$\tag{11}$$

where $*$ is the element-wise scaling operation.

Fully-connected ($FC$) layers parameterized by weight $W$ and bias $b$ can be easily modified to handle $\text{JVP}_{in}$ from the previous layer and has already been derived and used in prior works (Achille et al., 2021; Mu et al., 2020). We include it below for completeness.

$$FC_{lin}(x) = FC(x) + \Delta W^T x + \Delta b + W^T \, \text{JVP}_{in} \tag{12}$$

Non-linearities are also conveniently handled by the same technique. We illustrate the derivation for the GeLU activation commonly used in transformer-based networks:

$$GeLU_{lin}(x) = GeLU(x) + \left( \frac{GeLU(x)}{x} + x \cdot CDF(x) \right) \cdot \text{JVP}_{in} \tag{13}$$

where $CDF(x)$ evaluates the Standard Normal CDF at $x$. As before, all terms can be easily computed without any backpropagation steps.

Table 1: Comparison of non-linear fine-tuning (NLFT) vs TAFT on various downstream datasets sorted by distance to ImageNet (Achille et al., 2021). We compare fine-tuning the last attention block (NLFT-1, TAFT-1), the last 7 attention blocks (NLFT-7, TAFT-7), and only the classification head (FC). On most datasets sufficiently close to the ImageNet pre-training task, we show that TAFT can yield comparable or better performance compared to NLFT and FC, while benefiting from linearity.

| Dataset | NLFT-7 | TAFT-7 | NLFT-1 | TAFT-1 | FC |
|---|---|---|---|---|---|
| Caltech-256 | 93.7 | 95.7 | 95.8 | 95.9 | 95.7 |
| MIT-67 | 86.7 | 88.4 | 88.1 | 89.3 | 87.8 |
| Oxford Pets | 93.0 | 94.7 | 94.2 | 94.5 | 94.0 |
| Stanford Dogs | 84.8 | 91.6 | 91.2 | 91.9 | 90.7 |
| CUB-200 | 87.3 | 89.3 | 88.8 | 89.0 | 87.7 |
| FGVC-Aircrafts | 77.7 | 60.5 | 69.9 | 62.2 | 58.2 |
| Stanford Cars | 75.6 | 71.3 | 67.3 | 67.2 | 57.4 |
| Average | 85.5 | 84.5 | 85.0 | 84.3 | 81.6 |

The final linearized transformer, which we term Tangent Transformer, is simply the composition of such layers, chained together using the modified forward pass. Since Tangent Transformers are linear only in the weights $\Delta w$, and highly non-linear in the original weights $w$, we only update $\Delta w$ during fine-tuning, a process we term Tangent Attention Fine-Tuning (TAFT).

## 3.2 PARALLEL TRAINING AND COMPOSITION

Given $N$ models linearized about pre-trained weights $w$ and a query $x$, the ensemble of these models, defined by the linear combination of their outputs, is equivalent to evaluating a single tangent model composed by taking the same affine combination of component models in weight space:

$$\sum_{i=1}^{N} \lambda_i f_w^{lin\,i}(x) = f_w(x) + \nabla_w f_w(x) \cdot \sum_{i=1}^{N} \lambda_i \Delta w_i \qquad (14)$$

This gives rise to a natural interpretation of weight space composition via output ensembling, while reducing the cost of ensembling $N$ models from $\mathcal{O}(N)$ to $\mathcal{O}(1)$. In other words, we can train multiple Tangent Transformers on multiple different datasets in a completely parallel manner, and simply combine their output weights to yield a single model that performs as well as their ensemble but in constant inference time. Such compositions in weight space of transformer networks have also been previously explored by Wortsman et al. (2022b) combining multiple models trained on the same dataset with different configurations using weight averaging. However, as we show in Sec. 4.3, the lack of any theoretical relationship between combinations of models in weight space and their resulting output causes the resulting model to perform poorly when component models are trained on disjoint sets of data. On the other hand, we will show that the equivalence of weight averaging and ensembling allow the composition of up to 50 T-ViTs trained on different shards of data with relatively much smaller accuracy trade-offs compared to naively composing non-linear models.

## 3.3 ZERO-/LOW-COST FORGETTING WITH TANGENT TRANSFORMERS

"Learning" a model by combining the weights of component tangent models, each trained on disjoint shards of data, also allows for the subtraction of each component from the final model. Clearly, this subtraction operation completely removes the influence of samples contained within the shard used to train the component model from the final model. This is highly advantageous for machine unlearning.

Given a request to forget a training sample, the paragon unlearning method that guarantees forgetting of the target sample requires training the entire model from scratch on the remaining dataset samples. This is clearly impractical especially for large real-world transformer-based models like GPT-3 (Brown et al., 2020). With a Tangent Transformer composed from individual component models, we can simply remove the shard containing the sample to be forgotten by subtracting the weights of the associated component model. This theoretically guarantees forgetting while preserving accuracy when number of forgetting requests is small (Fig. 1(a)), all at essentially zero computational cost.

We note that this method of unlearning through shard removal is not scalable, since performance of the composed model degrades as number of forgetting requests increases. Instead, one can also

Table 2: We compose multiple T-ViTs, each trained with TAFT-1 on a disjoint shard of a dataset. The equivalence between linearly combining weights and output ensembling enables the composed T-ViT to outperform Model Soup (Wortsman et al., 2022b) across all datasets and sharding factors.

| Dataset / Shards | 10 Shards | | 25 Shards | | 50 Shards | |
|---|---|---|---|---|---|---|
| | Soup | TAFT | Soup | TAFT | Soup | TAFT |
| Caltech-256 | 94.5 | 95.0 | 93.2 | 94.3 | 90.8 | 93.3 |
| MIT-67 | 86.4 | 86.9 | 84.3 | 85.7 | 80.3 | 84.2 |
| Oxford Pets | 93.3 | 93.6 | 92.0 | 92.2 | 89.5 | 91.3 |
| Stanford Dogs | 90.9 | 91.4 | 90.4 | 90.9 | 88.8 | 90.4 |
| CUB-200 | 82.0 | 86.5 | 69.7 | 84.2 | 63.0 | 81.4 |
| FGVC-Aircrafts | 38.2 | 57.6 | 18.6 | 53.0 | 14.7 | 47.7 |
| Stanford Cars | 19.8 | 58.4 | 10.6 | 49.2 | 8.4 | 41.5 |
| Average | 72.2 | **81.3** | 65.5 | **78.5** | 62.2 | **75.7** |

optionally retrain the component model on the remaining samples in the shard, after removing the sample to be unlearned. Since shards are much smaller than the full dataset, this enables orders of magnitude speedup compared to the paragon of re-training from scratch, yet guarantees forgetting of the requested samples and maintains generalization performance of the resulting model (Fig. 1(c)).

### 3.4 TAFT WITH DIFFERENTIAL PRIVACY

Differential privacy (Dwork et al., 2014) is a mathematical framework to design algorithms which protect the privacy of individual training samples. Given a training dataset $D$, and an algorithm $M$, we say that $M$ is $(\epsilon, \delta)$-differentially private (DP) only if

$$P(M(D) \in E) \leq e^\epsilon P(M(D_{-i}) \in E) + \delta$$

for all $E$, $D_{-i}$, where $D_{-i}$ is obtained by removing the $i^{\text{th}}$ sample from $D$. In simple words, DP enforces an algorithm to produce similar outputs when the dataset differs by a single sample. One of the most popular ways of enforcing DP in deep learning is to use DP-SGD (Abadi et al., 2016) during training. DP-SGD introduces two modifications over the standard stochastic gradient descent (Robbins & Monro, 1951), first it clips the gradient norm of every sample, and then it adds gaussian noise to the sum of the clipped gradients across a training batch. Thus the information pertaining to individual samples is contained with clipping with noise perturbation. It is well known (Bassily et al., 2021; 2014; Fang et al., 2023; Yang et al., 2022) that convex models have better convergence and utility guarantees with trained differentially private convex optimization algorithms (in our case DP-SGD). We show in Tab. 1 that TAFT on Tangent Transformers provide comparable results to (in some cases better than) non-linear fine-tuning. As such, our experiments in Sec. 4.5 seek to understand if such models can remain effective in DP settings to reap the benefits of theoretical guarantees provided by private convex optimization.

### 3.5 CHOOSING A GOOD INITIALIZATION POINT

Strong pre-training objectives provide a natural initialization point at which we can compute the tangent model. However, linearizing transformer models around the full pre-training weights might exhibit strong feature biases towards the source pre-training dataset that might not transfer well to downstream tasks, especially in the later layers. As such, we propose a simple method to overcome this, by linearizing about a randomized re-initialization for the later attention layers, while keeping the pre-training weights constant for earlier layers in the network. We show that this significantly improves results in Fig. 2(b). For Vision Transformer-based classifiers, we further show in Fig. 2(c) that the CLS token itself can also be linearized in the same manner. We will empirically show that this can be beneficial for certain downstream tasks which are "far" from the pre-training initialization.

## 4 EXPERIMENTS

In Sec. 4.2, we show that TAFT on Tangent Transformers can attain similar performances on downstream tasks compared to non-linear fine-tuning. We show the advantages that arise from linearity for composition and parallel training in Sec. 4.3, machine unlearning in Sec. 4.4, and privacy

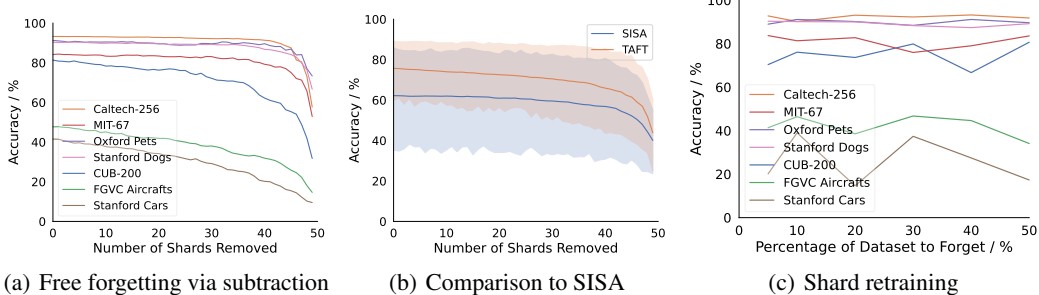

(a) Free forgetting via subtraction  (b) Comparison to SISA  (c) Shard retraining

Figure 1: **(a)** We show that when number of samples to forget is small, we can simply remove shards by subtracting the weights of their respective component model with minimal drop in final model accuracy (computed as an expectation over a uniform distribution of sample forgetting requests). **(b)** We compare against SISA (Bourtoule et al., 2021) which also uses a sharding technique for zero-cost unlearning. Our method is uniformly better across all number of shards removed on all datasets. **(c)** Retraining on the remaining samples in a shard after a forgetting request can further improve accuracy of the "unlearned" model, while enjoying up to $50\times$ faster training time compared to full re-training.

in Sec. 4.5. We describe our implementation details in Sec. 4.1, and carry out ablation studies on our implementation choices in Sec. 4.6. In Appendix C.4 and C.1, we also present ablations on different pre-training schemes, and comparisons against Linearized ResNets.

## 4.1 IMPLEMENTATION DETAILS

We run all our experiments on Vision Transformers on image classification tasks. In particular, we use ViT-L/16 (Dosovitskiy et al., 2020) as the base model in all our experiments, and linearize around its ImageNet pre-trained weights, the result of which we call T-ViT-L/16. We evaluate on the following datasets in increasing order of distance from the ImageNet pretraining task based on Li et al. (2020) - Caltech-256 (Griffin et al., 2007), MIT-67 (Quattoni & Torralba, 2009), Oxford Pets (Parkhi et al., 2012), Stanford Dogs (Khosla et al., 2011), CUB-200 (Wah et al., 2011), FGVC-Aircrafts (Maji et al., 2013), and Stanford Cars (Krause et al., 2013). Further details can be found in the Appendix.

## 4.2 HOW WELL DOES THE TANGENT TRANSFORMER COMPARE TO THE ORIGINAL MODEL?

While linearity yields many benefits in terms of composition, privacy, forgetting, and even interpretability, there is one main drawback - Tangent Transformers are strictly less expressive compared to the original non-linear model. Hence, for such linear models to be practical, we wish to preserve as much performance as possible on downstream tasks. We show in Tab. 1 that in fact, due to the strong inductive priors from the ImageNet pre-trained initialization, the average downstream performance is highly comparable with that of non-linear fine-tuning of the original model, differing on average only by $1.0\%$ and $0.7\%$ respectively when fine-tuning multiple attention blocks and just the last attention block. In fact, for several tasks that are close to the pre-training dataset (ImageNet) such as MIT-67, Stanford Dogs, Oxford Pets, and CUB-200, we show that TAFT actually outperforms non-linear fine-tuning. We hypothesize that this results from the implicit regularization imposed by the linearity constraints. We further note that for tasks that are far from the pre-training dataset, such as Stanford Cars and FGVC-Aircrafts, the local approximation becomes less accurate. As expected, the divergence between non-linear fine-tuning and TAFT increases. However, compared to transfer learning that simply fine-tunes the classification head, TAFT is strictly more expressive and hence improves by up to $2.9\%$ on average while maintaining linearity in weights.

Since most of the accuracy gains can be made from just fine-tuning the last attention block (NLFT-1, TAFT-1), this also allows for parameter-efficient fine-tuning where the number of parameters are $< 5\%$ of that needed for full fine-tuning. As such, we adopt NLFT-1/TAFT-1 in the following sections for non-linear/linear fine-tuning respectively, where we explore several benefits that linearity yields.

Table 3: DP fine-tuning of Tangent Transformers compared to regular non-linear fine-tuning. "Full" fine-tunes the entire (last) attention block, "BitFit" (Zaken et al., 2021) fine-tunes the bias parameters of the attention block, "Layer Norm" fine-tunes the affine parameters of layer normalization modules, and "FC" fine-tunes the classification head. "Ours" refer to fine-tuning the linearized parameters, and "NLFT" refers to fine-tuning the original parameters. Training tangent models outperforms its non-linear counterparts in all training regimes (*italics*: best for each regime, and **bold**: best overall)

| Dataset | Privacy | Full | | BitFit | | Layer Norm | | FC |
|---|---|---|---|---|---|---|---|---|
| | | Ours | NLFT | Ours | NLFT | Ours | NLFT | |
| CUB-200 | $\epsilon = 1$ | *46.3* | 11.0 | *44.5* | 41.9 | **46.7** | 45.9 | 44.9 |
| | $\epsilon = 3$ | 72.2 | 57.4 | *72.6* | 71.9 | **72.7** | 71.3 | 72.1 |
| | $\epsilon = 8$ | 82.2 | 77.2 | *82.0* | 81.7 | **82.4** | 82.1 | 82.1 |
| Stanford Cars | $\epsilon = 1$ | *4.3* | 1.4 | *4.9* | 4.8 | *4.6* | 4.8 | **5** |
| | $\epsilon = 3$ | *13.6* | 5.7 | **15.0** | 13.7 | *14.1* | 13.6 | 13.9 |
| | $\epsilon = 8$ | *26.8* | 16.6 | **27.5** | 26.3 | *27.2* | 27.0 | 26.2 |

## 4.3 COMPOSITIONALITY AND PARALLEL TRAINING

We evaluate our proposed method for parallel training and composition described in Sec. 3.2. We first shard a dataset into $N$ disjoint subsets, and train individual models on each subset. Note that training can be done in parallel, yielding a $N\times$ speed-up in training time. In Tab. 2, we show that across various sharding factors ($N = 10, 25, 50$) of each dataset, linearly combining weights of models fine-tuned with TAFT significantly outperforms composing separately trained non-linear models via Model Soup (Wortsman et al., 2022b), which to the best of our knowledge, is the only method that yields a composed model with $\mathcal{O}(1)$ inference cost (with respect to number of component models). Indeed, naively composing non-linear models through weight averaging yields no theoretical guarantees regarding how the output space of the resulting model changes. However, composing the linear weights of Tangent Transformers trained via TAFT is theoretically equivalent to output ensembling, hence outperforms Model Soup by 9.1%, 13.0%, and 13.5% on 10, 25, and 50 shards respectively, while maintaining an $\mathcal{O}(1)$ inference cost.

## 4.4 MACHINE UNLEARNING

Tangent Transformers composed from component tangent models trained on disjoint shards of data enables forgetting "for free", since unlearning can be done by simply subtracting models without needing any further training. We show in Fig. 1(a) that for a model composed from 50 shards, one can drop up to half of the shards (25) with only 4.0% drop in accuracy. We also compare against SISA (Bourtoule et al., 2021), which also drops shards upon forgetting requests, and show that we perform uniformly better across all datasets and number of shards dropped, and on average by 11.0%.

Optionally, one can retrain the shard containing the sample to be forgotten on the remaining samples in the shard. Even then, this still yields significant advantages compared to the baseline of re-training a model from scratch, since only the relevant shard needs to be retrained. In Fig. 1(c), we show that this yields a $50\times$ speed-up in our experiments, achieving close to the paragon performance (Appendix C.3) with only a 6.2% drop in accuracy after unlearning 50% of the entire dataset.

## 4.5 PRIVACY

We hypothesize that combining TAFT with differential privacy results in a better utility privacy trade-off resulting from convexity of the loss landscape. To illustrate this, we fine-tune various parameters of T-ViT-16 on two different fine-grained datasets (CUB200-easy and Stanford Cars-hard) for different privacy range. In Tab. 3, we observe that under almost all settings, privately fine-tuning the linearized parameters performs much better than privately fine-tuning the non-linear parameters. While fine-tuning the entire last attention block (column "Full" in Tab. 3) we observe that the gradient noise significantly degrades the model utility compared only fine-tuning the last fully-connected layer (and biases/normalization layers) of the network. The linear nature of tangent transformers along with the results in Tab. 3 also inspires a simple private composition/continual learning algorithm i.e. train private models on shards of data, and linearly combine their weights.

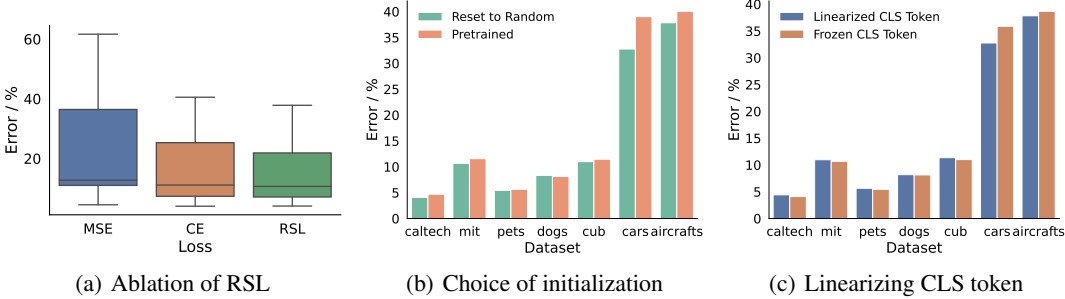

|  (a) Ablation of RSL | (b) Choice of initialization | (c) Linearizing CLS token |

Figure 2: **(a)** RSL can improve fine-tuning performance, beating CE and MSE by $1.5\%$ and $9.0\%$ respectively across 7 datasets. **(b)** While computing the tangent model about the full pre-training initialization is already effective on its own, re-initializing the weights of the last attention block before linearization can yield further performance gains. **(c)** Linearizing the CLS token improves accuracy on downstream datasets which are far from the pre-training tasks.

### 4.6 ABLATION STUDIES

We also conduct ablation studies to show key implementation details needed to train tangent models to perform comparable to non-linear models in downstream tasks. In particular, Fig. 2(a) shows that using the rescaled square loss significantly improves average-case results across all datasets by an average of $9.0\%$ and $1.5\%$, and on the hardest dataset by $23.8\%$ and $2.7\%$ compared to the MSE and CE loss respectively. In Fig. 2(b), we show that by resetting the weights of the final attention layer prior to linearization, average performance across datasets improves by $1.5\%$. We hypothesize that this is due to the negative transfer (Zhang et al., 2022) of later features learnt from the pre-training task to new downstream tasks. Indeed, we note that for datasets which are very close to ImageNet (*i.e.* Caltech-256, MIT-67), linearizing about the original pre-trained weights perform marginally better since they are highly transferrable to these downstream tasks. Similarly, we show in Fig. 2(c) that resetting and linearizing the CLS token in the last attention block of a vision transformer network can significantly improve performance on datasets which are far from the ImageNet pretraining task, improving results on FGVC-Aircrafts and Stanford Cars by $0.8\%$ and $3.1\%$ respectively.

## 5 DISCUSSION

Tangent Transformers linearized about a strong pre-trained point can serve to facilitate a number of processes related to fine-tuning and ensembling. Independently trained linear components can be easily composed, thus realizing full parallelism, and disgorged if need be, thus realizing deterministic removal of data.

However, linearization is not panacea: For the linearized models to work as advertised, the point around which the model is linearized is important, which can only be ascertained empirically. Once that is done, linear components can be trained with convex losses, which leads to overall models that enjoy strong guarantees for convergence, privacy, and composition. This limitation be further mitigated via techniques such as resetting certain pre-trained weights and linearization of the CLS token, as shown in our experiments. Another limitation of our method is that the inference cost of a Tangent Transformer can potentially be double that of the original model, since the modified forward pass requires an additional set of computations in addition to the original forward pass. However, we show that linearizing the last attention block of a ViT-L/16 model is often sufficient to yield strong performances on several downstream tasks. Under this regime, training and inference is parameter efficient, and linearization only incurs a slight increase in inference cost. Note that during training where the dataset is fixed, inference costs can actually be reduced to that of the original non-linear model by simply caching the activations from the static pre-trained weights for each training example and for each layer. Lastly, as observed by Koch & Soll, sharding naturally incurs significant trade-offs in performance on minority classes when training on highly imbalanced datasets.

The tasks on which we demonstrated how the linearity of transformers can be exploited through TAFT are certainly not exhaustive. Yet the encouraging empirical results of TAFT make it a candidate replacement for any applications of transfer learning or fine-tuning, while benefiting from the simplicity, composability, and interpretability of linear models.

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

# Supplementary Material

## A  IMPLEMENTATION DETAILS

We run all our experiments with ViT-L/16, and its tangent model termed T-ViT-L/16. Unless indicated otherwise, we adopt parameter-efficient fine-tuning by training only the last attention block of the vision transformer, along with the last normalization and fully connected layer. For experiments using TAFT in Table 1 and 2, and Figures 1(a)-(c), we train with the RSL loss using $\kappa = 15$. We also adopt a 30 epoch learning schedule for each dataset/task, with learning rate decay by a factor of 10 at epochs 15 and 25. We use a batch size of 32 for all our experiments, and train using Adam optimizer. We search over learning rates (LR) of $\{0.001, 0.0001\}$ for both non-linear fine-tuning and TAFT. We also select the best base model to linearize by performing TAFT on both the default pre-trained model, and resetting the last attention block and classification layer before linearization (PT=T and PT=F respectively), and ablating over resetting and linearizing the CLS token (CLS=T/F). We list the best configurations for non-linear fine-tuning (NLFT) and TAFT in Tab. 4, Tab. 5 (Table 1, main paper) and Tab. 6 (Table 2, main paper). We do not use any data augmentation for our experiments.

For experiments on composition and machine unlearning, datasets are split into muliple shards with respect to a fixed random seed by uniform sampling without replacement. For experiments on differential privacy, we use the standard cross entropy loss. For each $\epsilon = 1, 3, 8$, we use a 50 epoch training schedule with no learning rate decay and search over $PT \in \{T, F\}$. We predict only using the JVP output of the network. Gradients at each epoch are also aggregated over the entire dataset.

Table 4: Best Hyperparameters - Full Data, Single Attention Block

| Dataset | FC | NLFT-1 | LFT-1 |
|---------|-----|--------|-------|
| Caltech-256 | LR=1e-4 | LR=1e-4, CLS=F | LR=1e-3, $\kappa$=15, PT=F, CLS=F |
| MIT-67 | LR=1e-3 | LR=1e-4, CLS=F | LR=1e-3, $\kappa$=15, PT=F, CLS=F |
| Oxford Pets | LR=1e-3 | LR=1e-4, CLS=F | LR=1e-3, $\kappa$=15, PT=F, CLS=F |
| Stanford Dogs | LR=1e-4 | LR=1e-4, CLS=F | LR=1e-4, $\kappa$=15, PT=T, CLS=F |
| CUB200 | LR=1e-3 | LR=1e-4, CLS=T | LR=1e-3, $\kappa$=15, PT=F, CLS=F |
| Stanford Cars | LR=1e-3 | LR=1e-4, CLS=T | LR=1e-3, $\kappa$=15, PT=F, CLS=T |
| FGVC-Aircrafts | LR=1e-3 | LR=1e-4, CLS=T | LR=1e-3, $\kappa$=15, PT=F, CLS=T |

Table 5: Best Hyperparameters - Full Data, Last 7 Attention Blocks

| Dataset | NLFT-7 | LFT-7 |
|---------|--------|-------|
| Caltech-256 | LR=1e-4 | LR=1e-4,CLS=F,PT=T |
| MIT-67 | LR=1e-4 | LR=1e-4,CLS=F,PT=T |
| Oxford Pets | LR=1e-4 | LR=1e-4,CLS=F,PT=T |
| Stanford Dogs | LR=1e-4 | LR=1e-4,CLS=F,PT=T |
| CUB200 | LR=1e-4 | LR=1e-4,CLS=F,PT=T |
| Stanford Cars | LR=1e-4 | LR=1e-4,CLS=F,PT=T |
| FGVC-Aircrafts | LR=1e-4 | LR=1e-4,CLS=T,PT=T |

## B  DERIVATION OF LINEAR ATTENTION

We note that when taking the Taylor approximation for any (multivariable) function $f$, $f(w + \Delta w) = f(w) + \nabla_w f(w)^T \Delta w + \Delta w^T \nabla_w^2 f(w) \Delta w + \mathcal{O}(\|\Delta w\|^2)$ where $\mathcal{O}(\cdot)$ notation hides the higher order terms, the first order term can be efficiently computed via its directional derivative

$$\nabla_w f(w)^T \Delta w = \lim_{r \to 0} \frac{\partial f}{\partial r} f(w + r\Delta w)$$

where $r$ is a scalar variable. We will use this technique to derive the linearized closed form for the attention layer.

Table 6: Best Hyperparameters - Shards

| Dataset | NLFT-1 | LFT-1 |
|---|---|---|
| Caltech-256 | LR=1e-3 | LR=1e-3,PT=F,CLS=F |
| MIT-67 | LR=1e-3 | LR=1e-3,PT=F,CLS=F |
| Oxford Pets | LR=1e-3 | LR=1e-3,PT=F,CLS=F |
| Stanford Dogs | LR=1e-3 | LR=1e-3,PT=F,CLS=F |
| CUB200 | LR=1e-3 | LR=1e-3,PT=F,CLS=F |
| Stanford Cars | LR=1e-3 | LR=1e-3,PT=F,CLS=T |
| FGVC-Aircrafts | LR=1e-3 | LR=1e-3,PT=F,CLS=T |

Let $A$ denote the attention function parameterized by weights $W_q, W_k, W_v$. We wish to derive a closed form expression for the linear attention $A_{lin}$, which is defined as the first-order Taylor approximation of $A$ parameterized by the new linearized weights $\Delta W_q, \Delta W_k, \Delta W_v$.

$$A(x) = \Phi(x)V(x), \quad \text{where } \Phi(x) = \sigma(Q(x)K(x)^T), \tag{15}$$

$$Q(x) = \langle W_q, x \rangle, K(x) = \langle W_k, x \rangle, V(x) = \langle W_v, x \rangle \tag{16}$$

where $\sigma$ is the soft-max activation function. As in the main paper, we will write $Q, K, V$ instead of $Q(x), K(x), V(x)$ for ease of notation. We will derive the closed form for the single-headed attention, which can then be extended to multi-headed attention with minimal modification. Similarly, we will use $n = 1$ in the below proof (so $x$ is a vector in $\mathbb{R}^d$) for simplicity, but note that the final result extends to any $n > 1$.

$$A_{lin}(x) = A(x) + \lim_{r \to 0} \frac{\partial}{\partial r} A(x, W_q + r\Delta W_q, W_k + r\Delta W_k, W_v + r\Delta W_v) \tag{17}$$

$$= A(x) + \lim_{r \to 0} \underbrace{\sigma(\langle W_q + r\Delta W_q, x \rangle^T \langle W_k + r\Delta W_k, x \rangle) \langle W_v + r\Delta W_v, x \rangle}_{:=s} \tag{18}$$

Denote for ease of notation $\Delta Q = \langle \Delta W_q, x \rangle$, $\Delta K = \langle \Delta W_k, x \rangle$, $\Delta V = \langle \Delta W_v, x \rangle$. Then for each component $i$ of vector $s$, we can write

$$s_i = \sigma\left((Q + r\Delta Q)_i(K + r\Delta K)\right)^T (V + r\Delta V) \tag{19}$$

Applying chain rule, we get

$$\lim_{r \to 0} \frac{\partial}{\partial r} s_i$$

$$= \lim_{r \to 0} \left[ \sigma'\left((Q + r\Delta Q)_i (K + r\Delta K)\right) \left( \left( \Delta Q + W_q^T \frac{\partial x}{\partial r} \right)_i K + Q_i \left( \Delta K + W_k^T \frac{\partial x}{\partial r} \right) \right) \right]^T V$$

$$+ \lim_{r \to 0} \sigma\left((Q + r\Delta Q)_i(K + r\Delta K)\right)^T \left( \Delta V + W_v^T \lim_{r \to 0} \frac{\partial x}{\partial r} \right)$$

$$= \left[ \sigma'(Q_i K) \underbrace{\left( \left( \Delta Q + W_q^T \lim_{r \to 0} \frac{\partial x}{\partial r} \right)_i K + Q_i \left( \Delta K + W_k^T \lim_{r \to 0} \frac{\partial x}{\partial r} \right) \right)}_{:=\Psi_i} \right]^T V$$

$$+ \sigma(Q_i K)^T \underbrace{\left( \Delta V + W_v^T \lim_{r \to 0} \frac{\partial x}{\partial r} \right)}_{:=\Gamma}$$

$$= \left[ (\text{diag}(\sigma(Q_i K)) - \sigma(Q_i K)\sigma(Q_i K)^T)\Psi_i \right]^T V + \sigma(Q_i K)^T \Gamma$$

$$= \left[ \text{diag}(\Phi_i)\Psi_i - \Phi_i \Phi_i^T \Psi_i \right]^T V + \Phi_i^T \Gamma$$

$$= \left[ \Phi_i \odot \Psi_i - (\Phi_i^T \Psi_i)\Phi_i \right]^T V + \Phi_i^T \Gamma$$

where $\odot$ denote the Hadamard product. Hence, denoting $\Psi$ as the matrix with rows $\Psi_i$ and $\mathbb{1}$ the identity matrix, we obtain the desired result

$$A_{lin}(x) = A(x) + \lim_{r \to 0} \frac{\partial}{\partial r} s \tag{20}$$

$$= A(x) + \left( \Phi \odot \Psi - (\mathbb{1} \odot (\Phi^T \Psi)) \Phi \right)^T V + \Phi \Gamma \tag{21}$$

## C ADDITIONAL COMPARISONS

We discuss additional comparisons to Linearized ResNets in Sec. C.1, and detail training and inference times in Sec. C.2. We also compare our unlearning method with the paragon of re-training from scratch in Sec. C.3, and ablate on the pre-training schemes for initializing the tangent transformer in Sec. C.4.

### C.1 COMPARISON TO LINEARIZED RESNET ARCHITECTURES

The benefits of linearization rely on the strength of the inductive prior obtained from pre-training. Since vision transformers are shown to learn better inductive priors than convolutional architectures as the scale of training data increases, we believe that linearized transformers yield a clear advantage over linearized ResNets by being able to leverage the better inductive priors learnt from pre-training. We compare with linearized ResNet-50 in Tab. 7, where we show that TAFT outperforms Linearized ResNet-50 by 7.3% on the standard fine-tuning task, and by 9.0% for the parallel training and composition task (10 shards) averaged across 3 datasets.

Table 7: Comparing linearized ResNet-50 (L-RN50) and linearized ViT-L/16 (TAFT) on downstream classification tasks for both standard fine-tuning and parallel training/composition across 10 shards.

| Dataset | Shards | L-RN50 | TAFT |
|---|---|---|---|
| Caltech-256 | - | 85.5 | **95.9** |
| MIT-67 | - | 79.3 | **89.3** |
| Oxford Pets | - | 93.1 | **94.5** |
| Caltech-256 | 10 | 83.6 | **95.0** |
| MIT-67 | 10 | 72.4 | **86.9** |
| Oxford Pets | 10 | 92.5 | **93.6** |

### C.2 TRAINING AND INFERENCE TIME COMPARISONS

We compare the per-example training and inference wall-clock timings for NLFT and TAFT in Tab. 8. The inference and training cost for the linearized transformer is potentially twice of the original model as discussed in Sec. 5. We note that the train timings reported would be much faster in practice due to large batch sizes and caching of intermediate features when limiting training to later layers.

Table 8: Comparison of per-example training and inference wall-clock timing (seconds) for NLFT and TAFT using a batch size of 1. These would be much faster in practice due to large batch sizes and caching of intermediate features when limiting training to later layers. Timing is computed using the MIT-67 dataset.

| NLFT (Train) | TAFT (Train) | NLFT (Inference) | TAFT (Inference) |
|---|---|---|---|
| 0.147s | 0.204s | 0.021s | 0.065s |

### C.3 COMPARISON WITH FORGETTING PARAGON

In Fig. 3, we compare the shard re-training forgetting method using TAFT to the paragon of re-training from scratch. Both methods guarantee complete unlearning, but TAFT is able to achieve close-to-paragon performance while speeding up unlearning by up to 50x.

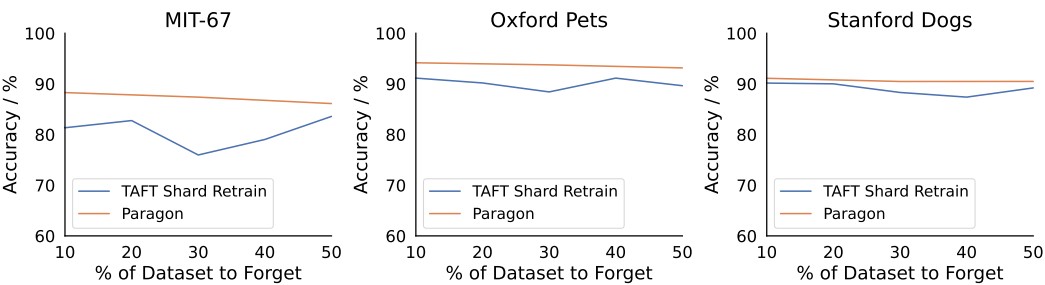

Figure 3: Shard re-training with TAFT (using sharding factor of 50) compared to the Paragon method of re-training the non-linear model from scratch. While both method guarantee complete unlearning, TAFT achieves close-to-paragon performance while speeding up unlearning by up to 50x.

## C.4 ABLATION ON PRE-TRAINING SCHEME

In practice, the choice of a model to fine-tune for downstream tasks presupposes some relation between the latter tasks and those used in the pre-trained initialization. Since we focus on classification, we choose ImageNet classification pre-training as our initialization for all the experiments.

Here, we compare TAFT with different pre-training schemes: (1) Self supervised learning via MAE (Masked Autoencoder Training), (2) Supervised/Classification pre-training, and (3) Contrastive Language-Image Pre-training (CLIP) followed by supervised pre-training.

We detail our results in Tab. 9. Indeed, the performance from fine-tuning depends on the discrepancy between the pre-training objective and the target task. (1) being the farthest to classification performs worse than a classification pre-training. However, by augmenting supervised classification pre-training using a contrastive language-image pre-training objective, (3) further boosts the performance of classification-only pre-training.

Table 9: We compare fine-tuning from three different pre-training schemes. (1) MAE does self-supervised pre-training via mask image modelling, (2) CLS uses ImageNet classification pre-training, and (3) CLIP uses contrastive language-image pre-training followed by fine-tuning on ImageNet classification. Since all MAE models are pre-trained on ImageNet 1K, we use the T-ViT-B architecture to fairly compare MAE and CLS where ImageNet 1K pre-trained models are available for both methods. The CLS T-ViT-L model is pre-trained on ImageNet 21K + 1K, while the CLIP model is pre-trained on WIT400M, ImageNet 12K + 1K. The inductive priors learnt from MAE transfer less effectively to the downstream classification tasks considered, where CLS on the smaller T-ViT-B model is able to outperform MAE on both T-ViT-B and T-ViT-L. The inductive priors learnt with CLIP, which combines both unsupervised contrastive learning and supervised fine-tuning, transfer best to the downstream tasks.

| Method | MAE / T-ViT-B | MAE / T-ViT-L | CLS / T-ViT-B | CLS / T-ViT-L | CLIP / T-ViT-L |
|---|---|---|---|---|---|
| MIT-67 | 67.5 | 75.4 | 77.7 | 89.3 | 91.8 |
| Oxford Pets | 78.8 | 90.0 | 91.7 | 94.5 | 95.1 |
| Stanford Dogs | 62.3 | 74.7 | 90.8 | 91.9 | 95.2 |

## C.5 INFLUENCE OF INDIVIDUAL COMPONENT MODELS

Since models are composed via linear combinations of their weights, the influence of a single component model can be quantified in at least two ways: (A) based on the difference in performance on a validation dataset when the component model is added, and (B) based on the magnitude of the difference in weights with and without the component model. We explored (A) in Fig. 1(a), where we show that subtracting models have lower impact on the performance on downstream tasks when the number of remaining component models is large. However when there remain only few component models in the composition, the impact of each model becomes larger.

In Fig. 4, we show that as a result of linearity, this effect is also reflected in the weight space via measuring the $L2$ difference in weights before and after adding the component model.

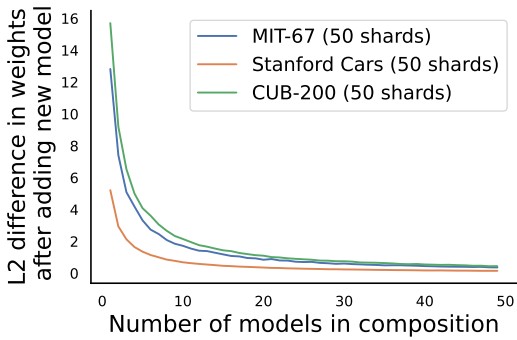

Figure 4: We plot the $L2$ change in weight space as a result of adding a new component model against number of existing models in the composition. The impact of adding a new model is significantly larger when number of existing component models is small. Note that while plotted on the same graph, the difference in scale between different datasets are not meant to be directly comparable due to difference in number of output classes, amongst other factors.

### C.6 TEXTURE CLASSIFICATION

In the main paper, we primarily evaluated our method on object classification tasks. In Tab. 10, we evaluate our method on the Describable Textures Dataset (DTD) (Cimpoi et al., 2014), where we show that even on texture classification tasks, composing models trained with TAFT consistently outperforms non-linear models across all sharding factors.

Table 10: We compare TAFT and Model Soup Wortsman et al. (2022b) in the same manner as Tab. 2 on the Describable Textures Dataset (DTD), and show that as a result of linearity, composing models trained with TAFT outperforms composing non-linear models across all sharding factors.

| Method | 10 Shards | 25 Shards | 50 Shards |
|---|---|---|---|
| Soup | 68.2 | 59.8 | 47.6 |
| TAFT | 75.3 | 71.6 | 65.3 |

### C.7 COMPARISON WITH PARAMETER-EFFICIENT FINE-TUNING

In this section, we compare against parameter-efficient fine-tuning methods. In particular, we compare against Adapters (Houlsby et al., 2019) and Low-Rank Adaptation (LoRA) (Hu et al., 2021) when applied to the same last attention block as non-linear fine-tuning and TAFT. Since the main use cases of such methods lie in parameter efficiency and training speed, we show in Tab. 11 that they typically exhibit lower performance on downstream tasks compared to full non-linear fine-tuning, and also lack the linearity of TAFT required to yield effective composition.

## D  TAFT WITH PROJECTED GRADIENT DESCENT

In the main paper, we constrain the distance that the $f_w^{lin}$ moves from its pretrained weights $w$ by using the L2 weight decay penalty as an regularizer during training, since the first-order taylor expansion is only valid around some local neighborhood of $w$. However, we note that it is also possible to impose a hard constraint rather than soft constraint using projected gradient descent, where weights are projected onto a ball of radius $R$.

In Tab. 12, we disable weight decay and instead train with projected gradient descent. We compare using the RSL loss (with $\kappa = 5$) and CE loss, since both losses differ in their effect on the final weight magnitude. We show that while RSL loss is more effective for the smaller radius $R = 1$,

Table 11: We compare TAFT with parameter-efficient fine-tuning methods – Adapter (Houlsby et al., 2019) and LoRA (Hu et al., 2021) – in the same manner as Tab. 2 on MIT-67, CUB-200, and Stanford Cars. We show that as a result of linearity, composing models trained with TAFT outperforms composing models fine-tuned with other methods across all sharding factors.

| Shards | Dataset | Adapter | LoRA | Soup | TAFT |
|--------|---------|---------|------|------|------|
| 10 | MIT-67 | 86.7 | 85.5 | 86.4 | 86.9 |
| | CUB-200 | 79.0 | 78.1 | 82.0 | 86.5 |
| | Stanford Cars | 20.3 | 21.4 | 19.8 | 58.4 |
| 25 | MIT-67 | 84.0 | 80.1 | 84.3 | 85.7 |
| | CUB-200 | 69.0 | 60.0 | 69.7 | 84.2 |
| | Stanford Cars | 14.7 | 9.6 | 10.6 | 49.2 |
| 50 | MIT-67 | 82.6 | 72.1 | 80.3 | 84.2 |
| | CUB-200 | 66.2 | 47.1 | 63.0 | 81.4 |
| | Stanford Cars | 12.0 | 5.2 | 8.4 | 41.5 |

Table 12: Projected Gradient Descent (PGD) onto ball of radius $R$. While imposing soft constraints through weight decay is more effective than PGD, the hard constraints on weight magnitude provide several benefits for bridging theoretical analysis and empirical results of deep neural networks.

| Dataset | $R$ | CE Loss | RSL Loss |
|---------|-----|---------|----------|
| Stanford cars | 1 | 37.6 | 52.1 |
| | 10 | 57.5 | 52.1 |
| CUB | 1 | 85.4 | 86.3 |
| | 10 | 88.4 | 86.4 |
| Aircrafts | 1 | 40.4 | 55.1 |
| | 10 | 58.5 | 55.4 |

CE loss becomes more effective as the radius increases to $R = 10$. While imposing such hard constraints generally yield worse results compared to TAFT, we note that this can be useful in several applications, such as for estimating the smoothness constant of the Tangent Transformer. This can help to better bridge the gap between theoretical analysis - which generally require $L$-smoothness assumptions or convex loss objectives - and empirical applications.

