# OpenReview forum: "Tangent Transformers for Composition,Privacy and Removal"
_ICLR.cc/2024/Conference — ICLR 2024 poster_

### Official Review · Reviewer_ZQ32 · 2023-10-29

**Soundness:** 3 good
**Presentation:** 3 good
**Contribution:** 3 good
**Rating:** 6
**Confidence:** 2

**Summary:**

The paper proposed a method for fine-tuning linearized transformers, called Tangent Attention Fine-Tuning (TAFT).

The paper claims that TAFT can achieve comparable performance to non-linear fine-tuning on various downstream tasks while enjoying the benefits of linearity such as compositionality, parallel training, machine unlearning, and differential privacy.

The paper also introduces Tangent Transformers, which are linearized versions of pre-trained transformer models, and shows how to compute their Jacobian-Vector products efficiently in a single forward pass.

The paper demonstrates the advantages of TAFT and Tangent Transformers in several experiments using vision transformer models.

To summarize, the paper's main contributions are:

- A novel method for fine-tuning linearized transformers that is computationally efficient and competitive with non-linear fine-tuning.

- A theoretical analysis of the benefits of linearity for model composition, parallel training, machine unlearning, and differential privacy.

- An empirical evaluation of TAFT and Tangent Transformers on various downstream visual classification tasks using vision transformer models.

**Strengths:**

The paper proposes a novel method for fine-tuning linearized transformers, called Tangent Attention Fine-Tuning (TAFT), which is computationally efficient and competitive with non-linear fine-tuning. The paper also introduces Tangent Transformers, which are linearized versions of pre-trained transformer models, and shows how to compute their Jacobian-Vector products efficiently in a single forward pass. The paper demonstrates the advantages of TAFT and Tangent Transformers on several experiments using vision transformer models.

In terms of originality, the paper introduces a new method for fine-tuning linearized transformers that is computationally efficient and competitive with non-linear fine-tuning. The paper also introduces Tangent Transformers, which are linearized versions of pre-trained transformer models, and shows how to compute their Jacobian-Vector products efficiently in a single forward pass. These contributions are novel and have not been explored before.

In terms of quality, the paper provides a theoretical analysis of the benefits of linearity for model composition, parallel training, machine unlearning, and differential privacy. The paper also provides an empirical evaluation of TAFT and Tangent Transformers on various downstream visual classification tasks using vision transformer models. The experiments are well-designed and the results are presented clearly.

In terms of clarity, the paper is well-written and easy to follow. The authors provide clear explanations of their methods and results. The paper also includes visualizations that help to illustrate the concepts presented.

In terms of significance, the paper’s contributions have important implications for the field of machine learning. The proposed method for fine-tuning linearized transformers is computationally efficient and competitive with non-linear fine-tuning. This has important implications for large-scale applications where computational efficiency is critical. Additionally, the theoretical analysis of the benefits of linearity has important implications for model composition, parallel training, machine unlearning, and differential privacy.

Overall, this is a well-written paper that makes significant contributions to the field of machine learning.

**Weaknesses:**

(1) While the paper is well-written and easy to follow, it would be helpful to provide more detailed explanations of some of the concepts presented. For example, the paper could provide more details on how Tangent Transformers are computed and how they are used in practice (especially from Eq 5 to Eq 6).

(2) While the paper provides an empirical evaluation of TAFT and Tangent Transformers on various downstream visual classification tasks using vision transformer models, it would be helpful to see more experiments that compare TAFT with other fine-tuning methods on these tasks. This would help to establish the competitiveness of TAFT more clearly.

(3) While the paper provides a theoretical analysis of the benefits of linearity for model composition, parallel training, machine unlearning, and differential privacy, it would be beneficial to provide more empirical evidence to support these claims. Specifically, it would be helpful to see more experiments that demonstrate the advantages of TAFT and Tangent Transformers on a wider range of tasks and datasets.
(e.g. DTD and UCF101).

Overall, this is a well-written paper that makes some contributions to the field. However, there is room for improvement in terms of providing more empirical evidence to support the claims made in the paper and providing more detailed explanations of some of the concepts presented.

**Questions:**

(1) Please give a detailed derivation process from Eq 5 to Eq 6 in the appendix.

(2) The selected datasets (Caltech-256 (Griffin et al., 2007), MIT-67 (Quattoni & Torralba, 2009), Oxford Pets (Parkhi et al.,
2012), Stanford Dogs (Khosla et al., 2011), CUB-200 (Wah et al., 2011), FGVC-Aircrafts (Maji
et al., 2013), and Stanford Cars (Krause et al., 2013)) are object-oriented. What about the performance of the proposed method on Describable Textures (DTD) (Cimpoi et al., 2014)？

(3) (Optional to reply) Could the proposed method be applied to temporal classification tasks, such as activity classification on UCF101?

---

> ### Author Response · Authors · 2023-11-14
> **Response to Reviewer ZQ32**
>
> We thank the reviewer for their thoughtful review of our paper. We address each concern in detail below.
>
> > **While the paper is well-written and easy to follow, it would be helpful to provide more detailed explanations of some of the concepts presented.**
>
> > **Please give a detailed derivation process from Eq 5 to Eq 6 in the appendix.**
>
> Thank you for the suggestion, we clarified some of the notation in Sec 3.1, and added the detailed derivation of the linearized attention function in Appendix B as suggested. We also plan to release code with the full implementation with the final version of the paper.
>
> > **While the paper provides an empirical evaluation of TAFT and Tangent Transformers on various downstream visual classification tasks using vision transformer models, it would be helpful to see more experiments that compare TAFT with other fine-tuning methods on these tasks. This would help to establish the competitiveness of TAFT more clearly.**
>
> Thank you for the suggestion. We initially compared only to the best performing fine-tuning method – non-linear fine-tuning – since parameter-efficiency is not necessary in our present application. Following the reviewer's suggestion, we added comparisons against adapter and low-rank fine-tuning in Appendix C.7, Table 11, and show that as a result of the trade-off in number of parameters, their performance on downstream tasks are consistently lower than that of non-linear fine-tuning and TAFT.
>
> > **Specifically, it would be helpful to see more experiments that demonstrate the advantages of TAFT and Tangent Transformers on a wider range of tasks and datasets. (e.g. DTD and UCF101).**
>
> > **What about the performance of the proposed method on Describable Textures (DTD) (Cimpoi et al., 2014)？**
>
> We believe that our experiments have provided sufficient evidence across 7 different datasets across multiple tasks - composition, unlearning, privacy, and standard fine-tuning. However, we agree that evaluating on non-object-oriented datasets can better elucidate the advantages of TAFT. As such, we have evaluated our proposed method on the DTD dataset in Appendix C.6 and Table 10, where we show TAFT convincingly outperforms the non-linear fine-tuning paragon.
>
> > **(Optional to reply) Could the proposed method be applied to temporal classification tasks, such as activity classification on UCF101?**
>
> While indeed interesting, extension to video-based tasks such as UCF101 is beyond the scope of our paper. However, we do not see an immediate reason why our method would not extend to temporal classification (as long as architecture remains transformer-style), and would love to see future work exploring this application.

---

### Official Review · Reviewer_QE2Q · 2023-10-31

**Soundness:** 3 good
**Presentation:** 3 good
**Contribution:** 3 good
**Rating:** 6
**Confidence:** 4

**Summary:**

The authors introduce Tangent Attention Fine-Tuning (TAFT) for fine-tuning linearized transformers. It can perform comparably with fine-tuning the original non-linear network in various downstream visual classification tasks. It enjoys several advantages compared to non-linear fine-tuning when it comes to model composition, parallel training, machine unlearning, and differential privacy.

**Strengths:**

The paper is the first work to propose an efficient method to linearize models in the Transformer family of architectures, which is meaningful. The paper is clearly written with many experiments.

**Weaknesses:**

1.since this is a fine-tuning method, please provide more fine-tuning methods for comparison (lora,adapter...) in table 3.
2.I wonder how TAFT works when applied to LLM? maybe some experiments can be added.
3.please derive in detail how to get the closed form expression in equation 5 and 6.

**Questions:**

see weaknesses.

---

> ### Author Response · Authors · 2023-11-14
> **Response to Reviewer QE2Q**
>
> We thank the reviewer for their constructive feedback and suggestions. We respond to each comment below.
>
> > **since this is a fine-tuning method, please provide more fine-tuning methods for comparison (lora,adapter...) in table 3**
>
> Since we are not constrained by parameter usage, we used the best fine-tuning method available for comparison - full non-linear fine-tuning. The strengths of the suggested methods lie in parameter efficiency and training speed, and comparing their performance on downstream tasks would not be completely fair for them. Furthermore, since Table 3 is meant to illustrate the advantages for differential privacy, we believe that comparing to the suggested fine-tuning methods in a more general application such as model composition can better address the reviewer’s concern. Following the reviewer’s suggestion, we added a comparison in Appendix C.7 where we show that non-linear fine-tuning and TAFT both significantly outperform parameter-efficient variants.
>
> > **I wonder how TAFT works when applied to LLM? maybe some experiments can be added.**
>
> While we recognize the potential of extending our work to large language models (LLMs), we believe it falls outside the scope of our current research, which focuses on the composition and decomposition of vision transformer networks. Nevertheless, we appreciate the reviewer's suggestion and agree that applying TAFT to LLMs and other modalities would be a valuable area of future research.
>
> > **please derive in detail how to get the closed form expression in equation 5 and 6.**
>
> Thank you for the suggestion, we added the derivation to Appendix B. Along with this, we will also release the code for TAFT with the final version of the paper.

---

### Official Review · Reviewer_Ef2Q · 2023-10-31

**Soundness:** 4 excellent
**Presentation:** 3 good
**Contribution:** 3 good
**Rating:** 6
**Confidence:** 4

**Summary:**

This paper introduces Tangent Attention Fine-Tuning (TAFT), a method for fine-tuning linearized transformers that are derived by computing a First-order Taylor Expansion around a pre-trained initialization. The key contributions of the paper are as follows:

Efficient Jacobian-Vector Product Calculation: The authors demonstrate that the Jacobian-Vector Product resulting from linearization can be efficiently computed in a single forward pass. This reduces the training and inference costs of linearized transformers to a similar order of magnitude as the original non-linear models, all while maintaining the same number of parameters.

TAFT presents an efficient and effective method for fine-tuning linearized transformers obtained through Taylor Expansion. It maintains performance parity with non-linear models while providing advantages related to model composition, training efficiency, unlearning, and privacy. This has significant implications for the practical use of transformers in various downstream tasks.

**Strengths:**

1. Efficient Jacobian-Vector Product Calculation: The authors demonstrate that the Jacobian-Vector Product resulting from linearization can be efficiently computed in a single forward pass. This reduces the training and inference costs of linearized transformers to a similar order of magnitude as the original non-linear models, all while maintaining the same number of parameters.

2. Comparable Performance: When applied to various downstream visual classification tasks, the Tangent Transformer fine-tuned with TAFT performs on par with fine-tuning the original non-linear network. This suggests that the linearized version is a viable alternative without compromising performance.

3. Convex Fine-Tuning Loss: The paper highlights that Tangent Transformers are linear concerning a new set of weights, resulting in a convex fine-tuning loss. This convexity offers several advantages over non-linear fine-tuning, particularly in terms of model composition, parallel training, machine unlearning, and differential privacy.

**Weaknesses:**

1. In Section 3.4, the author mentions basically the same things as in Section 2 Related work-Pravicy with no new theoretical analysis about differential privacy.

**Questions:**

1. How about interpretability? A detailed analysis of how a given training sample affects the learned model and the predicted results is given in LQF. Is it possible for the authors to provide an analysis of the interpretability?
2. The authors detail the advantages of linear models in the introduction, but these advantages don't seem to be relevant to transformer, what are the advantages and disadvantages of linearizing the transformer model compared to linearizing ResNet?

---

> ### Author Response · Authors · 2023-11-14
> **Response to Reviewer Ef2Q**
>
> We thank the reviewer for their valuable feedback and suggestions. We would like to provide a detailed response to each comment.
>
> > **In Section 3.4, the author mentions basically the same things as in Section 2 Related work-Privacy with no new theoretical analysis about differential privacy.**
>
> Section 2 provides comprehensive references to related work on differential privacy relevant to our paper. In Section 3.4, we establish a connection between differential privacy and TAFT. Existing theoretical works have proved (with tight theoretical bounds) that linear models (especially strongly convex, which TAFT is) enjoy much better utility guarantees compared to their non-linear counterparts when trained with DP-SGD. This makes TAFT an ideal candidate for DP fine-tuning, as it ensures that the model is differentially private along with strong utility guarantees. Since the theoretical results for strongly convex models are already tight, our focus is on developing such an architecture which can improve the utility of transformer models for challenging downstream tasks, such as fine-grained image classification.
>
> > **How about interpretability? A detailed analysis of how a given training sample affects the learned model and the predicted results is given in LQF. Is it possible for the authors to provide an analysis of the interpretability?**
>
> Thank you for the great suggestion! LQF requires a quadratic loss function to derive a closed form solution (via newton update) with and without the given training sample. This requires the computation of the inverse Hessian, which is often infeasible for large networks and requires approximation. Instead, our method of linearly composing disjoint component models to form the final model allows computing the influence of any particular component model (subset of the training samples) simply via measuring the difference in weights between the target component model and the composition (weight average) of the remaining models.
>
> As suggested, we added Appendix C.5, Figure 4, which demonstrates that as the number of model shards increase, the influence of each individual component model decreases. Due to linearity, this effect is also reflected in the model output space as seen in Figure 1(a).
>
> > **The authors detail the advantages of linear models in the introduction, but these advantages don't seem to be relevant to transformer, what are the advantages and disadvantages of linearizing the transformer model compared to linearizing ResNet?**
>
> Our comparison with linearized ResNet-50 (Appendix C.1, Table 7) demonstrates TAFT's superior performance, outperforming ResNet-50 by 7.3% and 9.0% for standard fine-tuning and composition tasks across three datasets, respectively. The advantages of linearization stem from the effectiveness of the inductive prior acquired during pre-training. Since vision transformers are shown to learn better inductive priors than convolutional architectures as the scale of training data increases, we believe that linearized transformers yield a clear advantage over linearized ResNets by being able to leverage the better inductive priors learnt from pre-training.
>
> Further investigation reveals that linearizing transformer models around the full pre-training weights can introduce strong feature biases towards the source pre-training dataset, potentially hindering transferability to downstream tasks, particularly in later layers (Figure 2(b)). Additionally, vision transformers often incorporate a [CLS] token for classification tasks. For downstream tasks significantly further from the pre-training initialization, we found that linearizing the [CLS] token itself can lead to improved performance (Figure 2(c)).

---

> > ### Comment · Reviewer_Ef2Q · 2023-11-22
> > **Excellent response**
> >
> > The author addressed all my questions point by point. After viewing other reviewer's comments and the author's responses, I think this paper deserves to be accepted.

---

> > > ### Author Response · Authors · 2023-11-22
> > > **Thank you**
> > >
> > > Thank you for your kind response and recommending acceptance for our work. We politely ask if the reviewer could kindly consider increasing their score to accept if they see fit. Thank you!

---

### Author Response · Authors · 2023-11-14
**Updated Rebuttal Revision  + New Experiments**

We thank the reviewers for their valuable feedback and suggestions. We have uploaded a revised draft with the following changes:

+ Added Appendix B to show full derivation for linear attention
+ Added Appendix C.5 and Figure 4 for experiments on influence of individual component models
+ Added Appendix C.6 and Table 10 for requested experiments on the DTD dataset
+ Added Appendix C.7 and Table 11 for comparisons with parameter-efficient fine-tuning methods
+ Minor changes to notation in Sec 3.1 to improve clarity

---

### Meta-Review · Area_Chair_tbaB · 2023-12-21

**Metareview:**

This paper looks at the problem of finetuning, specifically proposing a method for linearized transformer tuning derived through a Taylor Expansion around the pre-trained model. A theoretical argument is proposed as well as empirical results, demonstrating advantages in terms of model composition, efficiency, unlearning, and privacy.

 All of the reviewers appreciated the combination of theoretical motivation and empirical results, with a recommendation to accept. A few questions were raised in terms of interpretability, comparison to additional parameter-efficient fine-tuning methods, and additional experiments. Reviewers participating in the discussion mentioned that their concerns were addressed strongly, and therefore I recommend acceptance.

**Justification For Why Not Higher Score:**

Overall, the paper is solid, though results are limited to vision transformers at this time even thought it is a general fine-tuning method. However, I would not be opposed to bumping it up to spotlight given that finetuning is an important topic.

**Justification For Why Not Lower Score:**

None of the reviewers had major unaddressed concerns.

---

### Decision · Program_Chairs · 2024-01-16

Accept (poster)